# The Influence of Zero Shear Viscosity of TLA-Modified Binder and Mastic Composition on the Permanent Deformation Resistance of Mastic Asphalt Mixture

**DOI:** 10.3390/ma14185167

**Published:** 2021-09-09

**Authors:** Krzysztof Kołodziej, Lesław Bichajło, Tomasz Siwowski

**Affiliations:** Department of Roads and Bridges, Faculty of Civil and Environmental Engineering and Architecture, Rzeszow University of Technology, 35-959 Rzeszow, Poland; krzych@prz.edu.pl (K.K.); leszbich@prz.edu.pl (L.B.)

**Keywords:** mastic asphalt, TLA-modified binder, zero shear viscosity, filler-binder ratio, permanent deformation, static indentation test, dynamic indentation test

## Abstract

Mastic asphalt (MA) has been particularly popular in recent years for bridge pavements due to many advantages such as easy application, good waterproofing properties, and high durability. However, the drawback of mastic asphalt in comparison to other asphalt mixtures is its lower resistance to permanent deformation. Trinidad Lake Asphalt (TLA) is often applied to make mastic asphalt resistant to permanent deformation. Practical experience demonstrates that serious failures may occur if MA pavement design and materials selection is not taken into account sufficiently. Therefore in this study, the influence of two parameters: zero shear viscosity (ZSV) of TLA-modified binder and mastic composition described by the filler–binder ratio, on the permanent deformation resistance of the MA mixture was evaluated. The primary purpose of determining the ZSV of the TLA-modified binders was to evaluate the rutting potential of the binders. The permanent deformation (rutting) resistance of the MA mixtures was evaluated based on static and dynamic indentation tests. The optimum content of TLA in the base bitumen and the optimum filler–binder ratio in the MA mixture were obtained based on multiple performance evaluations for modified binder, mastic and MA mixtures, i.e., 20% and 4.0, respectively.

## 1. Introduction

Mastic asphalt (MA) has been particularly popular in recent years for bridge pavements due to many advantages such as easy application, good waterproofing properties, and high durability. Mastic asphalt commonly comprises a blend of penetration grade bitumen and usually a modifier (termed “modified binder”), which is mixed with fine aggregate (termed “filler”) to form “mastic”. The addition of coarse aggregate completes the composition of the mastic asphalt. Due to its very good fluidity and self-levelling performance when paving, mastic asphalt does not require compaction. The application of one or two layers of mastic asphalt in bridge pavement provides very good water and de-icing agents protection of the bridge deck due to the low permeability of the MA mixture with air void content below 1%. In comparison with standard asphalt concrete (AC) or stone mastic asphalt (SMA) pavements, mastic asphalt pavement has longer service life as well as greater fatigue life resulting from a high content of bitumen in the MA mixture, which is very viscoelastic and does not crack easily [1,2,3,4].

The drawback of mastic asphalt in comparison to other asphalt mixtures is its lower resistance to permanent deformation. The main concern with MA bridge pavement courses is the possibility of occurrence of the rutting (viscoplastic deformation), resulting from high service loads at high temperatures, as the underlying rigid deck contributes to an increase in stresses in the pavement and, consequently, accelerates rutting process [5,6,7]. Rutting has been distinguished as a primary distress mechanism and a major design criterion as long as MA pavements have been used. The rutting susceptibility of pavement is mainly influenced by aggregate and mixture properties. However, the characteristics of the binder are also important, especially for modified binders, which are claimed to improve the rutting resistance and extend pavement service life.

To make mastic asphalt resistant to permanent deformation, the products that “harden” the bitumen are used such as waxes, polymers and natural asphalts. In the latter case, Trinidad Lake Asphalt (TLA) is often applied. TLA is well recognized as an efficient bitumen modifier for its high compatibility, stability, and durability, thus it is often used in mastic asphalt for bridge pavements. The most important advantages of using TLA as a bitumen modifier include optimization of structural properties of the binder (lower brittleness and greater resistance to ageing), improvement of binder adhesion to mineral material, significant improvement of workability and compaction of the mixture, and ultimately, its higher resistance to permanent deformation and fatigue [8]. High-quality MA pavement with the TLA-modified binder has been successfully applied for many years on hundreds of bridges worldwide [9,10,11,12].

Bitumen rheology is a major factor influencing the permanent deformation of an asphalt mixture. Various rheological parameters have been used as a performance indicator for rutting. The rutting properties of an asphalt mixture are greatly affected by the characteristics of the modified binder and the relative proportions of binder and mineral fillers [13,14]. In this study, the zero shear viscosity (ZSV) of TLA-modified binders was used as a measure of permanent deformation resistance of mastic asphalt mixture. The concept of ZSV was introduced by Sybilski [15] as a suitable indicator to evaluate the partial contribution of the binder to the rutting resistance of bitumen pavements. The ZSV is said to be an indicator of two rutting-related binder characteristics, namely, the stiffness of the binder and the binder’s resistance to permanent deformation under long-term loading [16,17]. Henceforward, ZSV is being given a lot of attention by many researchers as a possible measure for the rutting resistance of modified binders [18,19,20,21].

The ZSV of a binder is an intrinsic property that has been shown to correlate with the rutting performance of the binder. However, within an asphalt mixture, it is probably more appropriate to consider the performance of the mastic (binder plus filler, i.e., fine mineral material) rather than simply considering the pure bitumen. The main role of mastic in the asphalt mixtures is to bind coarser aggregate grains and to prevent their segregation. The most basic parameter that controls the properties of mastic is its composition, i.e., the correct choice of proportion between binder and filler content [22,23,24]. Binder content in the mastic cannot be too low, since it would lead to excessive stiffness and brittleness, which, in effect, renders the pavement susceptible to cracking. Nonetheless, with a gradual increase in binder content, the effect of sliding or lubrication of aggregate particles intensifies, leading to a decrease in resistance to permanent deformation of the pavement. An increase in filler content leads to a stiffening effect, which increases the resistance to permanent deformation. The views on the optimum filler–binder ratio in various asphalt mixtures have changed over the years. Initially, a range of 0.6–1.2 was advised, regardless of the type of the mixture. Currently, the recommendations state that the ratio should not exceed 1.6 in coarse-graded mixtures and 1.4 in fine-graded mixtures [25,26]. The composition of mastic plays a vital role in terms of the high resistance of mastic asphalt pavements to permanent deformation [5]. Therefore, identification of optimum mastic composition (i.e., filler–binder ratio) may be the basis for the preliminary design of a mastic asphalt mixture [11].

Most existing studies focused on the characterization and improvement of engineering properties of MA mixtures. Little study has researched or provided the detailed design of MA mixtures such as the material selection, modifier content (if any), and binder-aggregate ratio. However, practical experience demonstrates that serious failures may occur, if MA pavement design and materials selection is not taken into account sufficiently. Therefore in this study, the influence of two parameters: zero shear viscosity of TLA-modified binder, which is directly dependent on the modifier content, and mastic composition described by the filler–binder ratio, on the permanent deformation resistance of mastic asphalt mixture were evaluated. The objective of this study is to evaluate a design for MA mixtures produced from the 35/50 base bitumen and three TLA dosages (0, 10, and 20%). The primary purpose of determining the ZSV of the TLA-modified binders was to evaluate the rutting potential of the binders. The filler–binder ratio in the range from 3.2 to 4.0 (which corresponds to the binder content in the MA mixture in the range from 6.9 to 8.7%, respectively) was considered in the tests. The optimum content of TLA in the base bitumen and the optimum filler–binder ratio in the MA mixture were obtained based on multiple performance evaluations for modified binder, mastic, and MA mixtures. The permanent deformation (rutting) resistance of mastic asphalt mixtures was evaluated based on static and dynamic indentation tests. The MA properties related to rutting were observed for short-term aged binders and in the upper range of pavement service temperatures.

## 2. Materials

Since the 35/50 penetration grade bitumen (PKN Orlen, Plock, Poland) is the only one recommended for MA bridge pavements by the Polish road administration, it was chosen as a base bitumen in this study. Moreover, the 35/50 bitumen is also the most popular one in Poland to be used in other types of road and bridge pavements. Table 1 lists some of the basic properties of the 35/50 bitumen in original (unaged) condition, determined according to the relevant European standards and compared with the European requirements. The TLA modifier used in this study was Trynidad Epure TE Z 0/8 (CMS Polska, Gdansk, Poland), whose basic properties are listed in Table 2.

As shown in the relevant tables, the base bitumen used in this study fulfilled the respective European requirements. However, in the case of TLA, small deviations from the requirements were revealed. This is due to the fact that the natural asphalt supplied by the manufacturer was covered by diatomite (an agent preventing asphalt pieces from sticking together) and was added to the mixer in this form. It was assumed in this study that the tests would be conducted on the binder delivered in the form available for the manufacturer. This agent could stiffen the TLA resulting in deviations revealed.

Limestone powder, produced following the relevant requirements [34,35], was used as a filler. The fraction of <0.063 mm was sieved from the limestone powder and used in mastic testing since the filler–binder ratio assumed in this study is the proportion by weight of particles passing through the 0.063 mm sieve to the total binder weight in the mixture. The density of the filler equaled 2.70 Mg/m^3^ and its air void content in a dry and compacted state was 31%. Granodiorite all-in aggregate of fraction 0/4 and coarse aggregate of fractions 4/8 and 8/11 were produced following the European standard [35], and used to complete a final mastic asphalt mixture.

The mastic composition, i.e., filler–binder ratio, was determined based on the typical compositions of mastic asphalt mixtures given in the Polish standard [36] since the newer requirements [34] do not include information on the maximum binder content in the MA mixture. Finally, the filler–binder ratios assumed in this study ranged from 2.0 to 4.0, with a step of 0.4 [37]. Since the TLA-modified binder is a mixture of bitumen and fine mineral particles, the quantity of the filler (limestone powder) was decreased by the number of mineral particles coming from the TLA, following the approach described in the handbook [38].

The rutting tests were performed on the MA 11 mixture, required for the higher traffic categories by the Polish road administration and most often applied in bridge pavements in Poland. The grading curve of the MA 11 mixture is presented in Figure 1. The MA11 mixtures prepared for testing included mastic with the filler–binder ratios of 3.2, 3.6, and 4.0. Choice of these ratios was based on the requirements set in [39] regarding both the minimum binder content in mastic asphalt (*B_min_* = 7.0%) and the maximum binder content that still provides satisfactory quality of the MA mixture. The binder contents in the tested MA mixtures were based on the above assumptions made for the filler–binder ratio and ranged from 6.9% (the value close to the minimum content *B_min_*), by 7.7% to the maximum value of 8.7% for the filler–binder ratios of 4.0, 3.6, and 3.2, respectively. The former authors’ tests revealed, that the greater binder content than 8.7% led to the production of a very soft mixture with static indentation results over 15 mm.

## 3. Methods

### 3.1. Preparation of TLA-Modified Binders

To achieve an effective and optimum modification, it is necessary to precisely establish the quantity of the modifier dosed into the base bitumen. To determine the optimum addition of the TLA, the base 35/50 bitumen was modified with TLA content from 0 to 20% with a step of 10% (relative to the weight of the base bitumen). Such initial TLA contents were based on the available studies [40,41,42,43] and the authors’ research [44,45], which implied that the addition of more than 20% of TLA could lead to excessive stiffness of the binder. Moreover, due to the high cost of the modifier, the chosen TLA content was optimized also based on the life-cycle cost analysis (LCCA) of the MA bridge pavement. Finally, three TLA-modified binders with the 35/50 base bitumen were selected in this study to evaluate the permanent deformation resistance of MA pavement.

The TLA-modified binders were prepared according to [46]. To obtain a homogeneous TLA-modified binder, the laboratory blender (IKA, Staufen im Breisgau, Germany) was used to ensure a constant mixing speed and thus no voids were created in the mixture. The base bitumen and TLA were heated until they became fluid before mixing. The base bitumen was preheated at 160 °C in an oven (Wamed, Warsaw, Poland) for 0.5 h to make it ready to mix. The modifier, a specified amount of TLA, was added into liquid base bitumen with the external addition method. The temperature of the modified binders was kept at 160 °C for 1 h, as recommended by the 35/50 bitumen manufacturer. After this time, the mixture was stirred at 3000 rpm for 5 min, so that the TLA was homogeneously dispersed in the base bitumen. Immediately after preparation, the ready mixture was poured into short-term ageing containers.

### 3.2. Binders RTFOT Ageing Procedure

Since short-term ageing has a significant influence on the resistance of hot mix asphalt to permanent deformation, while long-term ageing affects the resistance to fatigue, which is not important for MA pavements [43,47,48], only short-term ageing was considered in this study. The rolling thin film oven test (RTFOT) was used to simulate the short-term ageing. The RTFOT was conducted on the base bitumen and TLA-modified binders following [48]. This method consists of exposing a thin layer of binder to hot air for 75 ± 1 min. Glass bottles with the samples of the binder are placed in a special disc that rotates at 15.0 ± 0.2 rpm located in a laboratory oven, where hot air was periodically injected inside at a rate of 4.0 ± 0.2 l/min. The samples of binder in bottles were subjected to a temperature of 163 ± 1 °C.

After short-term ageing, the binder was poured into a large container, homogenized by mixing, and then poured into molds for appropriate tests. Efforts were made to minimize additional heating of the binder after simulated short-term ageing in order not to age it additionally. The physical properties tests were performed immediately after the ageing procedure. Before the rheological properties tests, the samples after ageing were poured into molds, left for 24 h, and then tested.

### 3.3. Binders Physical Properties Tests

Conventional tests of penetration at 25 °C and ring and ball (R&B) softening point were carried out to study the physical properties of the 35/50 base bitumen as well as the TLA-modified binders after ageing. The penetration test is a common test performed to characterize the hardness of bitumen and binders [28]. The R&B softening point test was carried out to determine the temperature at which a phase change in the binder occurs [29]. For both physical properties tests, the Grubbs test was performed to detect outliers in a univariate data set assumed to come from a normally distributed population.

### 3.4. Binders Rheological Properties Test

A theoretical concept of ZSV is a measure of the viscosity of a material when shear stress is acting on it at a shear rate of almost zero. Following the preparation of the modified binders, the dynamic shear rheological (DSR) test was conducted using sinusoidal oscillation loading. To calculate the ZSV of the binders, the Cross model was fitted to measure the complex viscosity (η*). This model allows extrapolating the complex viscosity at a zero frequency condition with nonlinear regression analysis.

The oscillatory tests using a HAAKE RheoStress 1 Rheometer (Thermo Scientific, Karlsruhe, Germany) were performed under strain-controlled loading conditions by applying a sinusoidal angular displacement of constant amplitude [49]. The test geometry with a 25 mm diameter plate and 1 mm gap was used. A constant strain of 5% was applied at a frequency between 0.01 and 100 Hz and in a temperature range from 40 to 60 °C with an increment of 10 °C. All tests were carried out in the linear viscoelastic range. The tests were performed on smooth plates, they were not additionally roughened. Before testing, the plates were heated to a temperature of 70–80 °C to soften the binder and ensure good adhesion to the plates. After proper preparation of the sample, it was cooled down to the test temperature. Before the test, the linear viscoelastic interval was determined. The deformation level was selected so that the sample in the entire range of tested frequencies was within this range.

The Cross model was used to estimate zero shear viscosity of the modified binders [50]. The Cross model describes flow curves of the modified binder in the form of a four-parameter function as follows:(1)$$\frac{\eta^{*}-\eta_{\infty }^{*}}{\eta_{0}^{*}-\eta_{\infty }^{*}}=\frac{1}{1+{\left(K\omega \right)}^{m}},$$
where:η* is the complex viscosity;η_0_* is the ZSV;η_∞_* is the limiting viscosity in the second Newtonian region;ω is the angular frequency [rad/s];K and m are constants.

The Cross model material parameters (η_0_*, η_∞_*, K, m) were determined with the HAAKE RheoWin 4.41 (Thermo Scientific, Karlsruhe, Germany), a complete measuring and evaluation software package, used to control the rheometer and to handle the measured data with the selected evaluation method. The Grubbs test was carried out to detect outliers in a univariate data set assumed to come from a normally distributed population.

### 3.5. Preparation of Mastic Asphalt Mixture

The binders were prepared earlier by mixing the 35/50 base bitumen with the appropriate amount of TLA modifier (0, 10, and 20%) according to [51]. It was followed by the preparation of the aggregate mixture, which comprised 45% coarse aggregate, 27% all-in aggregate, and 28% filler. As mentioned previously, three binder contents were chosen: 6.9, 7.7, and 8.7%, which corresponded to the filler–binder ratio of 4.0, 3.6, and 3.2, respectively. Next, the binder and the aggregate mixture were heated in an oven at the respective temperature (binder: 170 °C, aggregate: 230 °C) for 1 and 3 h, respectively. After this time, the heated components were mixed for 4–5 min, until all aggregate particles were coated by the binder. Then the final mastic asphalt mixture was placed in an oven again and stored at a constant temperature of 220 °C for 1 h. Finally, the warmed up MA mixture was poured into a steel cubic mold (dimensions: 70 mm × 70 mm × 70 mm) or cylindrical mold (diameter 150 mm, height 60 mm) and compacted to form test specimens. After cooling to room temperature, the specimens were demolded and prepared for the indentation tests.

### 3.6. Static and Dynamic Indentation Tests of Mastic Asphalt

The permanent deformation resistance of mastic asphalt mixtures was evaluated using static and dynamic indentation methods. Indentation tests were performed on nine mastic asphalt mixtures, prepared with three different binders having different TLA quantities from 0% by 10% to 20% and three different mastics having the filler–binder ratios of 3.2, 3.6, and 4.0.

Static indentation was tested using 70.7 mm cubic specimens formed at the temperature of 220 °C according to [52]. The load was transferred onto specimens using an indentor pin with a circular base (a cylindrical piston with a contact surface area of 500 mm^2^) and the penetration of the indentor into the specimen (indentation) was measured. The load was applied for some time starting with the initial loading of 25 N and then, after 10 min of preloading, increasing the load by 500 N, thus giving the total test load of 525 N. During the test, the specimen remained submerged in water at the constant temperature of 40 °C. The values registered as the results of the test included indentation of the piston after 30 min and the increase in the indentation after additional 30 min of constant loading.

Dynamic indentation tests were performed on cylindrical specimens with a diameter of 150 mm and height of 60 mm formed at the temperature of 220 °C according to [53,54]. Before testing, the specimens were conditioned for 4 h at the constant test temperature of 50 °C and then preloaded with a static load of 10 kPa. The load was transferred onto specimens using a cylindrical piston with a contact surface area of 2500 mm^2^ and a flat-ended base diameter of 56.4 mm. The parameters of cyclic loading were as follows:maximum load: 0.875 kN (corresponding to specimen’s stress of 0.35 N/mm^2^);minimum load: 0.2 kN (corresponding to specimen’s stress of 0.08 N/mm^2^);load time: 0.2 s;rest time: 1.5 s;duration of the cycle: 1.7 s;the shape of the loading curve: half-sine.

The results were recorded after 2500 and 5000 cycles of dynamic loading. The indentation of the piston into the specimen was determined as the average of three displacement values: two from linear variable differential transformer (LVDT) transducers (IPC Global, Victoria, Australia) placed on a plate mounted on the piston and one from the displacement sensor of the piston itself. The values registered as the results of the test included indentation of the piston after 2500 cycles and the increase in the indentation after additional 2500 (5000 total) cycles of dynamic loading.

In both static and dynamic indentation tests, six measurements were performed for each of the nine MA mixtures. As in the case of binder test results, before their further processing, outliers were excluded using the Grubbs test and measurement uncertainties were evaluated using the A method based on a statistical analysis of a series of repeated measurements.

## 4. Results and Discussion

### 4.1. Binders Properties

TLA has low penetration and high softening point due to the high content of ash and asphaltenes, so the addition of TLA would have a remarkable influence on the properties of the base bitumen. The physical properties determined for the TLA-modified binders after RTFOT ageing are presented in Table 3 and Figure 2. Figure 2a presents a substantial reduction in penetration value with the increasing TLA content. When the TLA content is up to 20%, the penetration of the RTFOT-aged binder decreases rapidly. It means the addition of TLA can slightly worsen the short-term ageing performance of TLA-modified binders. Figure 2b shows that the softening point in RTFOT-aged conditions corresponds to TLA content. The softening point of modified binders linearly increases with increasing TLA content.

ZSV determined for the base bitumen and TLA-modified binders are presented in Table 4 and Figure 3 depending on test temperature (40, 50, and 60 °C). The proportional decrease of ZSV is observed with an increase in temperature and a decrease in TLA content (Figure 3a). At 40 °C, the two-fold ZSV increase was obtained for 10% TLA content, and slightly less than the three-fold increase for 20% TLA as compared to the base bitumen. At the remaining temperatures, the relevant ZSV increments are smaller: 1.7 and 2.5 times at 50 °C and 1.6 and 2.4 times at 60 °C, for 10 and 20% TLA, respectively. Figure 3b confirms the linear decrease of the ZSV observed as the test temperature rises.

The additional results of a comprehensive laboratory investigation on the physical and rheological properties of the 35/50 base bitumen modified by the addition of three different TLA contents (0, 10, and 20% by weight) as well as on the effect of short-term ageing on these physical and rheological characteristics were presented by the authors elsewhere [55,56].

### 4.2. Static Indentation Results

As mentioned above, the MA mixture’s resistance to permanent deformation is determined by static (I) and dynamic (ET) indentations. The static indentation dependency on the ZSV of the TLA-modified binder and filler–binder ratio (f/b) is shown in Figure 4. This 3-D plot revealed that the static indentation can be credibly estimated using these two parameters, i.e., ZSV and filler–binder ratio. It can be seen that the mastic composition (f/b) has the higher influence on the static indentation of MA mixture than the binder rheological properties (ZSV). Furthermore, as the ZSV is related to the TLA content (Figure 3b), one can easily predict how the optimal quantity of this modifier should be dosed into the base bitumen. Since the static indentation can be used as a measure of the permanent deformation resistance of the MA mixture, choosing the specific parameters of the binder by the TLA content and/or the relevant mastic composition, the permanent deformation resistance of MA could be improved.

The mastic composition has a big impact on the static indentation of the MA mixture. The permanent deformation resistance directly depends on the mastic’s stiffness, since the coarse aggregate grains are suspended in the mastic and have no direct contact with each other. Increasing the filler–binder from 3.2 to 4.0 results in a decrease in static indentation by about half (Figure 5). On the other hand, with a constant value of the filler–binder ratio, the binder change by the TLA content increase no longer causes such large changes in indentation values. The decrease in indentation value at the constant ZSV value of the binder, which represents the MA mixture modification without the change of the binder, is similar for each level of the TLA content, here represented by the relevant ZSV value.

The increase in the ZSV value of TLA-modified binder, as an effect of the TLA addition, is most pronounced in a soft mixture, i.e., with a lower filler–binder ratio (Figure 6). In the case of a stiffer mixture—where the filler–binder ratio is 4.0—the static indentation decrease rate is much smaller. This difference in absolute terms is 2.42 and 0.84 mm for filler–binder ratios of 3.2 and 4.0, respectively. For percentages, this decrease rate is at a comparable level of 32 and 27%, respectively.

A parameter that can also be used to evaluate the MA resistance to permanent deformations is the increment of the static indentation (*Inc*) between 30 and 60 min of the test. The static indentation shows the resistance of the MA mixture at only one point, i.e., after a certain time, whereas the static indentation increment illustrates these changes over time. Mixtures with a smaller increment will be more resistant to permanent deformations, as the rut will develop more slowly than with mixtures with a larger increment. The dependence of increment of MA static indentation on ZSV of TLA-modified binder and the filler–binder ratio is shown in Figure 7.

It can be observed in Figure 8, that for a mixture without TLA (i.e., with the lowest ZSV), a change in the filler–binder ratio from 3.6 to 4.0 results in a more than three times decrease in the static indentation increment, while for a mixture with the 20% TLA addition (i.e., the highest ZSV) this decrease is about 2.5 times. It can also be noted that the use of the TLA modifier makes the most sense in the case of soft mixtures, i.e., where the filler–binder ratio is the smallest. An increase in ZSV results in a significant decrease in deformation growth (a decrease of almost 50%) compared to the mixture with a filler–binder ratio of 4.0, where this decrease is half as small (Figure 9). Although the TLA addition does not significantly improve resistance to permanent deformation, it very well limits the growth of ruts in the case of MA mixtures, where the binder determines the mixture properties.

The relationship between the rheological binder parameter (ZSV), the mastic composition, and MA static indentation, i.e., MA resistance to permanent deformations, can be described by a linear equation in the form of:(2)$$I=a_{1}\mathrm{ZSV}+a_{2}\left(f/b\right)+I_{0},$$
where:I—static indentation [mm];$a_{1},{\;a}_{2}$—constants determined based on regression analysis;ZSV—zero shear viscosity [Pa·s];(f/b)—filler–binder ratio [-];$I_{0}$—free constant.

For the assumed test conditions (RTFOT-aged binder, dynamic oscillatory ZSV test, static indentation test temperature 40 °C), the values of the following coefficients in the linear Equation (2) were obtained:$a_{1}=-1.524\times 10^{-6}$;$a_{2}=-4.587$;$I_{0}=21.993$.

The Equation (2) with the above coefficients can be used to estimate the static indentation of MA mixture and thus the permanent deformation resistance of MA pavement, starting from the choice of TLA content and determination of the zero shear viscosity of a TLA-modified binder and followed by choosing the relevant filler–binder ratio.

### 4.3. Dynamic Indentation Test

The differences in dynamic indentation values for the tested MA mixtures are greater than static indentation values (Figure 10). It may be also noticed that there is a much larger range of test results of dynamic indentation than with static indentation. It follows that the composition of the MA mixture has a greater impact on dynamic indentation values than static indentation ones. This is evident both in the case of the influence of the mastic composition as well as the influence of the rheological binder parameters, in this case depending on the amount of TLA. In the latter case, this effect is more pronounced for soft mixtures with a low filler–binder ratio.

In general, the dependence of MA dynamic indentation showed in Figure 10 is similar to static indentation tests (Figure 4). Only in the case of a mixture with 20% TLA content and the filler–binder ratio of 3.6, the MA mixture behaved differently from expectations, as a dynamic indentation result of 8.76 mm was achieved. In this test, both increasing the filler–binder ratio, as well as increasing the ZSV value by the addition of TLA caused the mixture to harden, which is seen in the decrease in dynamic indentation value (Figure 11 and Figure 12). The difference of almost 14 mm was obtained between the extreme results compared to just over 5 mm in static indentation tests, which confirms a much larger range of dynamic indentation test results.

Similar observations can be made from the increment of dynamic indentation evaluation (Figure 13). The addition of TLA reduces the increase of dynamic indentation (i.e., deformation over time) regardless of the filler–binder ratio. The reduction of the increment of dynamic indentation is almost 50% for the filler–binder ratio of 3.2 and less than 40% for the filler–binder ratio of 4.0 (Figure 14). This is a significant difference as compared to the static indentation test, in which for the ratio of 3.2 the addition of TLA reduced the increment of static indentation at a level similar to the dynamic test (approximately 48%), while for the ratio of 4.0 the reduction of the increment of static indentation gained 30% (Figure 15).

The relationship between the rheological binder parameter (ZSV), the mastic composition, and MA dynamic indentation, i.e., MA resistance to permanent deformations, can be described by a quadratic equation in the form of:(3)$$\mathrm{ET}=a_{1}{\mathrm{ZSV}}^{2}+a_{2}\mathrm{ZSV}+a_{3}{\left(f/b\right)}^{2}+a_{4}\left(f/b\right)+{\mathrm{ET}}_{0},$$
where:ET—dynamic indentation [mm];$a_{1},{\;a}_{2},{\;a}_{3},{\;a}_{4}$—constants determined based on regression analysis;ZSV—zero shear viscosity [Pa·s];(f/b)—filler–binder ratio [-];${\mathrm{ET}}_{0}$—free constant.

For the assumed test conditions (RTFOT-aged binder, dynamic oscillatory ZSV test, dynamic indentation test temperature 50 °C), the following coefficient values in the quadratic Equation (3) were obtained:$a_{1}=1.515\times 10^{-9}$;$a_{2}=-3.558\times 10^{-4}$;$a_{3}=-7.479$;$a_{4}=45.083$;${\mathrm{ET}}_{0}=-37.766$.

Similarly, as in the case of static indentation, Equation (3) with the above coefficients can be used to estimate the dynamic indentation of MA mixture and thus the permanent deformation resistance of MA pavement, starting from the choice of TLA content and determination of the zero shear viscosity of a TLA-modified binder and follow by choosing the relevant filler–binder ratio.

### 4.4. Correlation of Static and Dynamic Indentation

For assumptions made in static and dynamic indentation tests, the fitting of curves given by Equations (2) and (3) to the obtained test results was checked by regression analysis. The results are given in Table 5. Analysis of the correlation between the permanent deformation resistance parameters and the ZSV parameter indicates a high level of agreement of the ZSV parameter distribution and the penetration parameters, i.e., static and dynamic indentation. The result of fitting ZSV to the static indentation is about 95%, the slightly worse relationship can be observed using the regression results for the dynamic indentation (90%). It is worth emphasizing that the authors carried out similar estimations for other test conditions: the TLA-modified binders in the original (unaged) state, creep test of ZSV instead of oscillatory test and three temperatures of ZSV test in the range of 40–60 °C [55,56]. The analysis revealed that not one of these considered test conditions had a significant influence on the static as well as dynamic indentation test results.

There is a very good correlation between both methods of indentation testing (or testing of permanent deformation resistance) of mastic asphalt mixtures. The results of the dynamic indentation test are linearly correlated with the results for static indentation (Figure 16).

The relationship between the static and dynamic indentation can be described by a linear equation in the form of:(4)ET=1.510×I+1.913,
where:ET—dynamic indentation [mm];I—static indentation [mm].

Regression analysis taking into account all the results of the study showed that the coefficient of determination R^2^ is equal to 0.681. Figure 16 shows, however, that two observations differ slightly from the others, which may affect the level of equation fit to the obtained data. Therefore, the data set was subjected to a more detailed statistical analysis including the determination of the DFFITS parameter and Cook’s distance, which are a measure of the influence of the test values on the obtained regression equation. For the case under consideration, DFFITS should not exceed the value of 1, while Cook’s distance should be less than 4/18 = 0.222. For pairs (I; ET) equal to (4.40; 15.6) and (7.44; 18.75), it was found that the DFFITS parameters are 1.116 and 0.976, and in the case of Cook’s distance, these values are 0.273 and 0.483, respectively. In the case of the remaining results, both estimated parameters were below the threshold values. On this basis, it was decided to exclude these points from the regression analysis. After discarding the questionable results, the coefficient of determination R^2^ increased to 0.820. Thus, the obtained Equation (4) was very well suited to the analyzed test data.

### 4.5. Discussion

The test results revealed that ZSV of the binder is a very good measure to evaluate the permanent deformation resistance of the MA mixture. The high coefficients of determination R^2^ for both relationships (2) and (3) showed the high influence of the ZSV on the static and dynamic indentation. Increasing the viscosity of the binder by adding the TLA modifier positively affects the permanent deformation resistance of the MA mixture. The MA resistance improvement is visible in both the direct deformation value as well as the increase of deformation over time.

Evaluating the influence of ZSV on permanent deformation resistance, one should take into account the composition of the MA mixture. The positive effects of higher ZSV values are most visible in a mixture with a low f/b ratio, where the binder has more impact on deformation resistance. When the f/b ratio increases by increasing the filler content, and thus the MA mixture is stiff enough, the influence of ZSV on permanent deformation resistance is not visible so much. Regardless of the composition, the positive effect of the modified binder is revealed and the increase in binders’ viscosity by the TLA addition improves the deformation properties of the MA mixture. However, the degree of improvement depends on the MA composition.

In the case of both static and dynamic indentation tests, the minimum ZSV value can be indicated, above which there is a significant change in the permanent deformation resistance of the MA mixture. There are the ZSV values of 700 and 800 kPas for the temperatures of 40 and 50 °C, respectively. In the case of the static indentation test, it is visible in the increase of static indentation, (particularly when the f/b ratio equals 3.2) and less visible in static indentation itself. In the case of the dynamic indentation test, the improvement of permanent deformation resistance is visible in both dynamic indentations itself as well as in the increase of dynamic indentation.

When comparing the two test methods, it can be concluded that the dynamic method shows greater differences in permanent deformation resistance testing compared to the static method. In the static indentation test, the total relative difference for MA mixtures with the extreme values of the TLA additions is only 32 and 27% for the mastic composition f/b of 3.2 and 4.0, respectively. In the dynamic indentation test, the respective differences are 56 and 40%. The greater sensitivity of the dynamic method to changes in the composition of the MA mixture was revealed, particularly in terms of binder type and proportions between components.

## 5. Conclusions

A comprehensive laboratory investigation on the influence of zero shear viscosity of a TLA-modified binder and mastic composition on the permanent deformation (rutting) resistance of mastic asphalt pavement was carried out and presented in the paper. The 35/50 base bitumen modified by the addition of three different TLA contents (0, 10, and 20% by weight) was used as a binder. The filler–binder ratio in the range from 3.2 to 4.0 (which corresponds to the binder content in the MA mixture in the range from 6.9 to 8.7%, respectively) was considered in the tests. The static and dynamic indentation tests of the respective MA mixtures were carried out to evaluate their permanent deformation resistance. Based on the test results of this study, the following conclusions can be made:the ZSV parameter should be used as an indicator describing permanent deformation of mastic asphalt mixtures in the rutting test; this is warranted in particular when the MA mixtures are modified with TLA modifiers;the use of TLA to modify the 35/50 base bitumen significantly improves the ZSV level, thus effectively increasing the resistance of the mastic asphalt mixture to permanent deformation;zero shear viscosity of TLA-modified binders shows high correlation with the parameters of permanent deformation, i.e., static and dynamic indentations of MA mixtures, for the assumed test conditions, i.e., RTFOT-aged binder, dynamic oscillatory ZSV test, test temperatures of 40 and 50 °C for static and dynamic indentation tests, respectively;the minimum ZSV value can be indicated, above which the significant improvement of the permanent deformation resistance of the MA mixture was revealed in terms of both static and dynamic indentations and their increases; there are the ZSV values of 700 and 800 kPas for the temperatures of 40 and 50 °C, respectively;the Equations (2) and (3) can be used for pre-determination of the MA composition based on ZSV value of TLA-modified binder; for the assumed filler–binder ratio one can estimate the required TLA content so that the MA mixture meets the requirements of the permanent deformation resistance; the optimum content of TLA in the base bitumen and the optimum filler–binder ratio in the MA mixture were obtained in the study within the ranges of 10–20% and 3.6–4.0, respectively;the dynamic indentation method (as compared to the static method) is more sensitive to changes in the composition of the MA mixture and the binder type; however, there is a very good correlation between static and dynamic methods–the dynamic indentation values are on average 1.5 times larger than the static ones.

## Figures and Tables

**Figure 1 materials-14-05167-f001:**
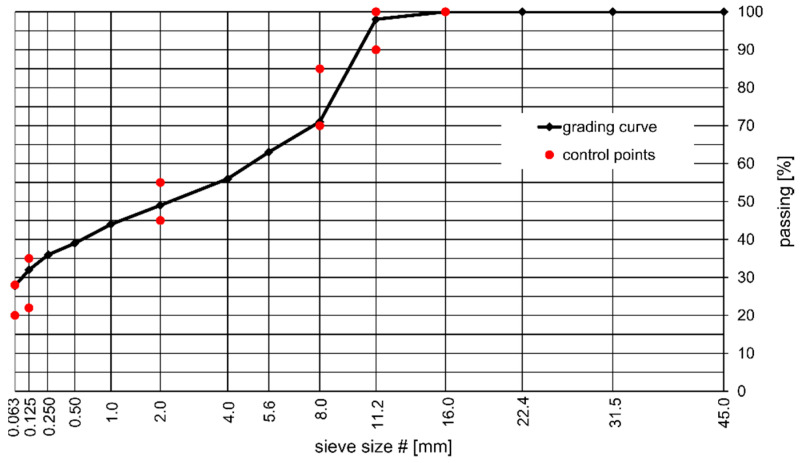
Grading curve of the tested MA 11 mixture.

**Figure 2 materials-14-05167-f002:**
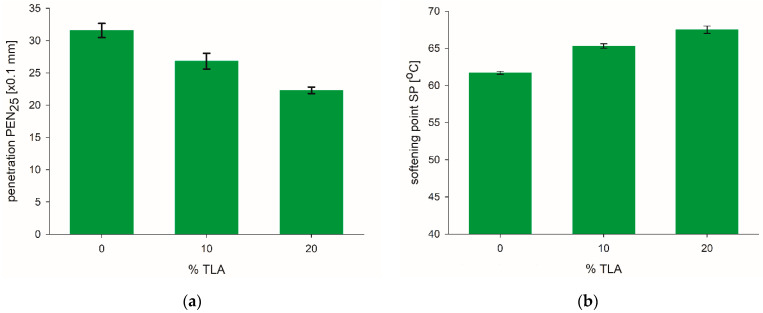
(**a**) Penetration and (**b**) softening point after RTFOT ageing of TLA-modified binders with different TLA content.

**Figure 3 materials-14-05167-f003:**
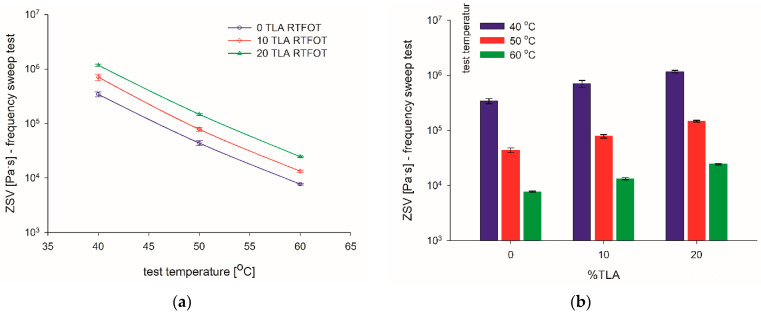
ZSV values of tested binders for: (**a**) various test temperatures and (**b**) TLA content.

**Figure 4 materials-14-05167-f004:**
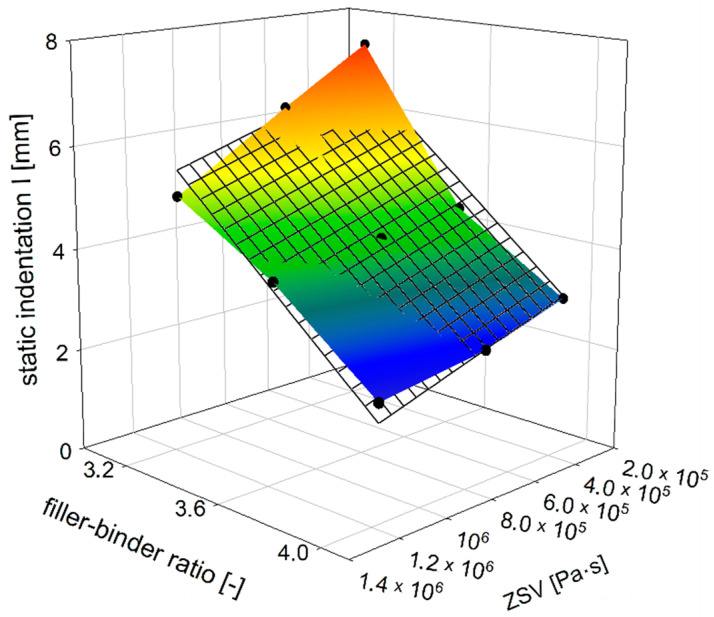
The dependence of MA static indentation on ZSV of TLA-modified binder and filler–binder ratio.

**Figure 5 materials-14-05167-f005:**
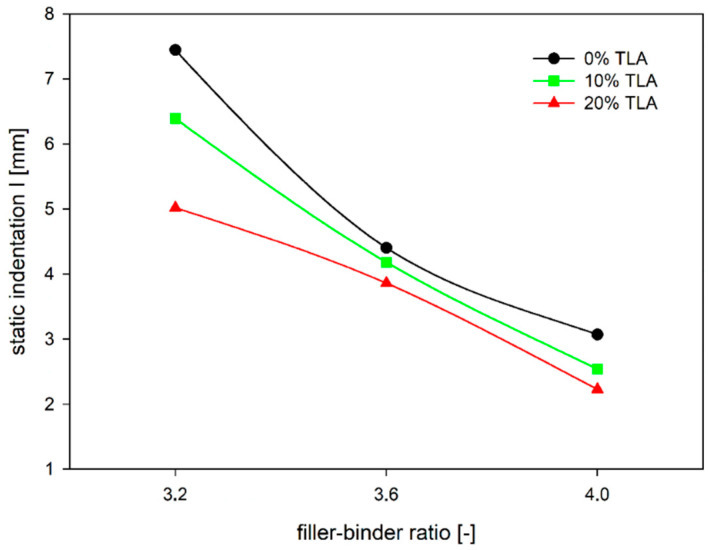
The dependence of MA static penetration on the filler–binder ratio for a different level of TLA addition.

**Figure 6 materials-14-05167-f006:**
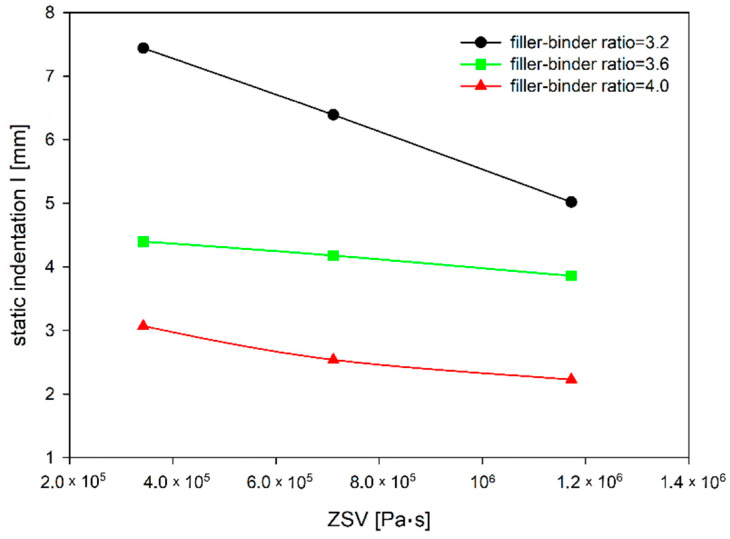
The dependence of MA static indentation on ZSV of TLA-modified binder for different filler–binder ratios.

**Figure 7 materials-14-05167-f007:**
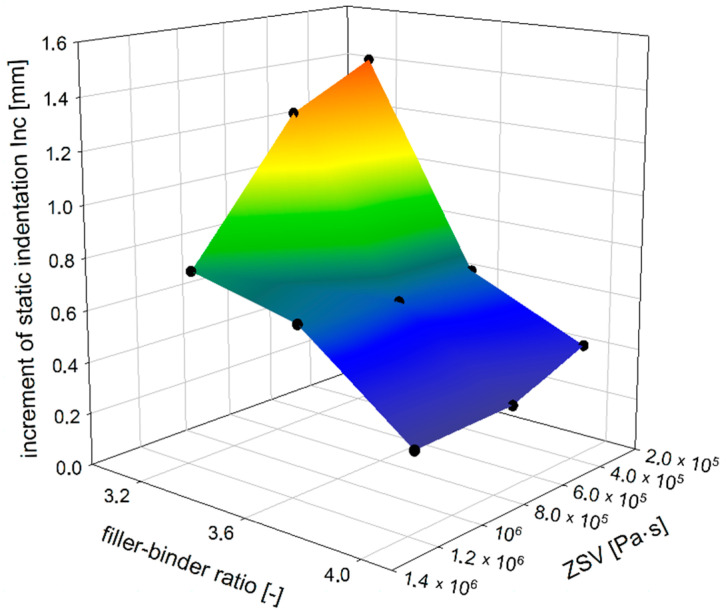
The dependence of increment of MA static indentation on ZSV of TLA-modified binder and filler–binder ratio.

**Figure 8 materials-14-05167-f008:**
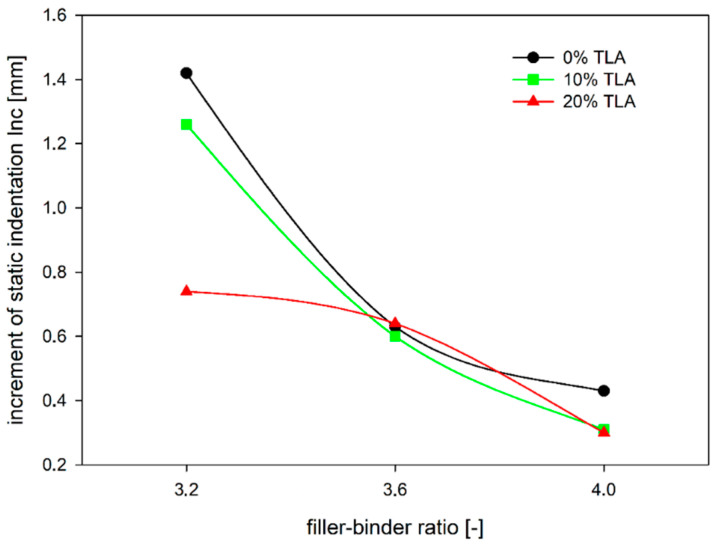
The dependence of increment of MA static indentation on the filler–binder ratio for different TLA contents.

**Figure 9 materials-14-05167-f009:**
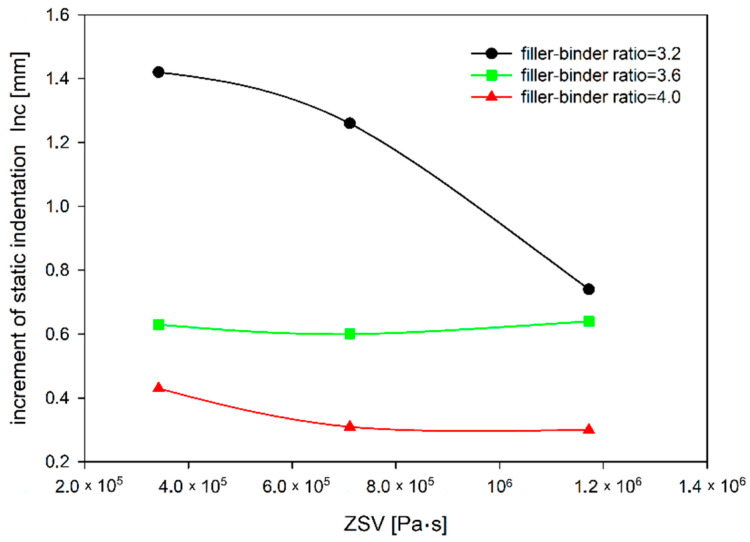
The dependence of increment of MA static indentation on ZSV of TLA-modified binder for different filler–binder ratios.

**Figure 10 materials-14-05167-f010:**
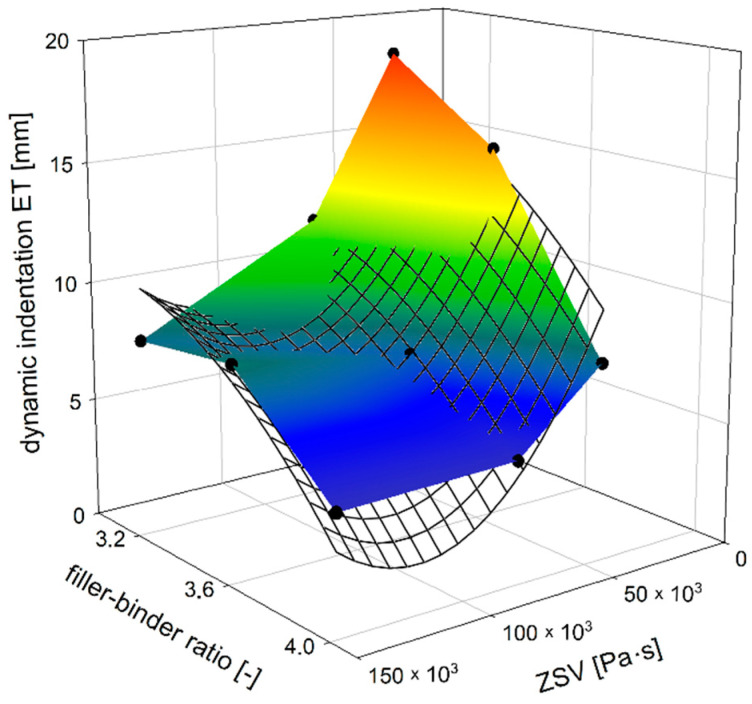
The dependence of MA dynamic indentation on ZSV of TLA-modified binder and filler–binder ratio.

**Figure 11 materials-14-05167-f011:**
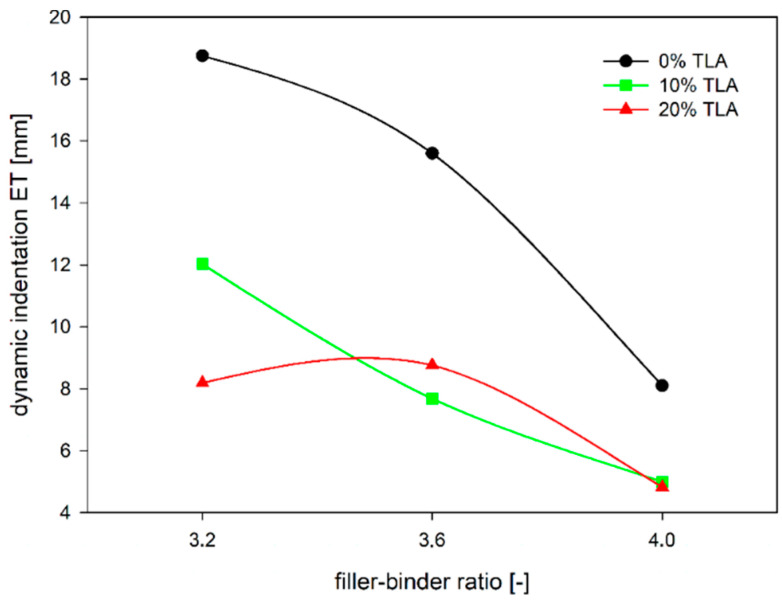
The dependence of dynamic indentation on the filler–binder ratio for different TLA contents.

**Figure 12 materials-14-05167-f012:**
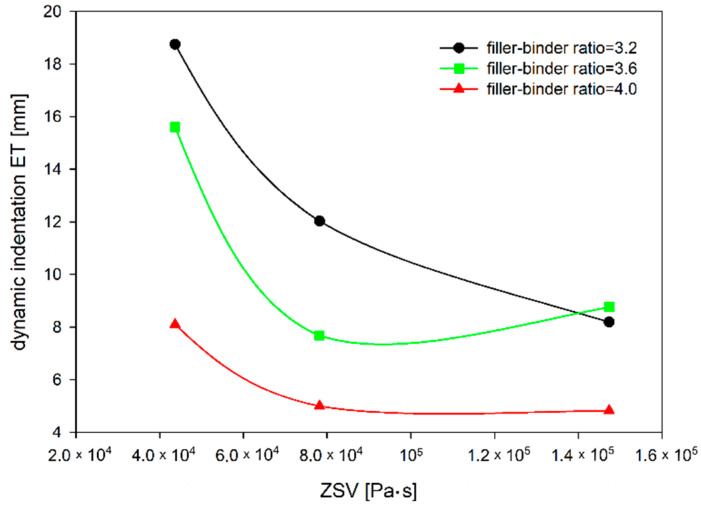
The dependence of dynamic indentation on ZSV of TLA-modified binder for different filler–binder ratios.

**Figure 13 materials-14-05167-f013:**
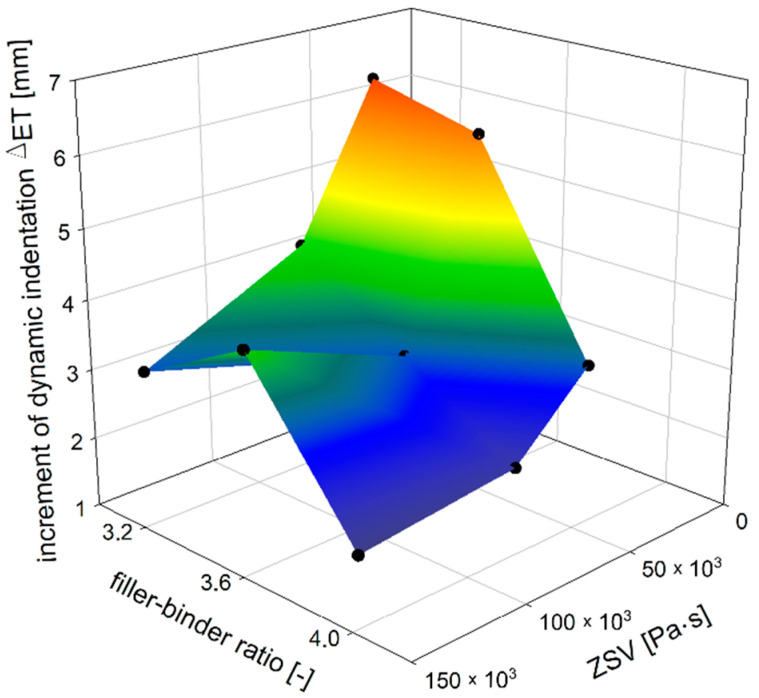
The dependence of increment of MA dynamic indentation on ZSV of TLA-modified binder and filler–binder ratio.

**Figure 14 materials-14-05167-f014:**
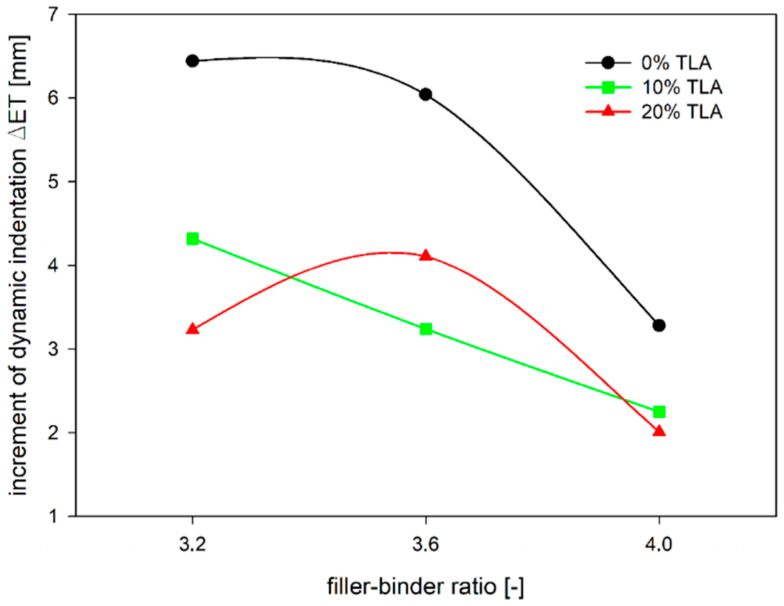
The dependence of increment of MA dynamic indentation on the filler–binder ratio for different TLA addition.

**Figure 15 materials-14-05167-f015:**
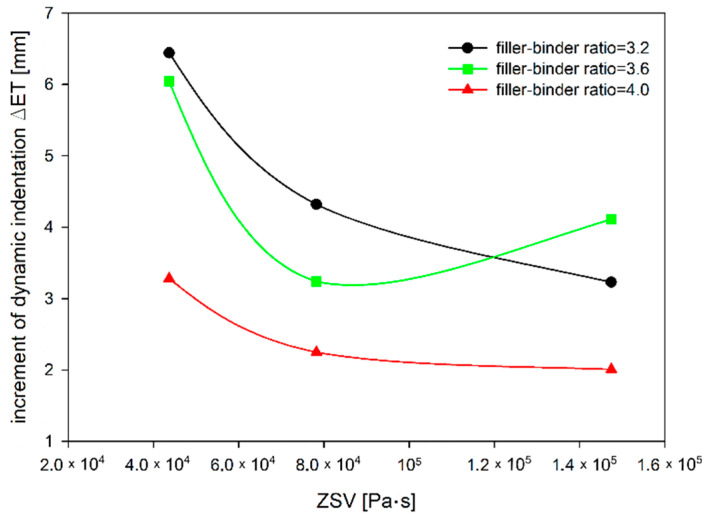
The dependence of increment of MA dynamic indentation on ZSV of TLA-modified binder for different filler–binder ratios.

**Figure 16 materials-14-05167-f016:**
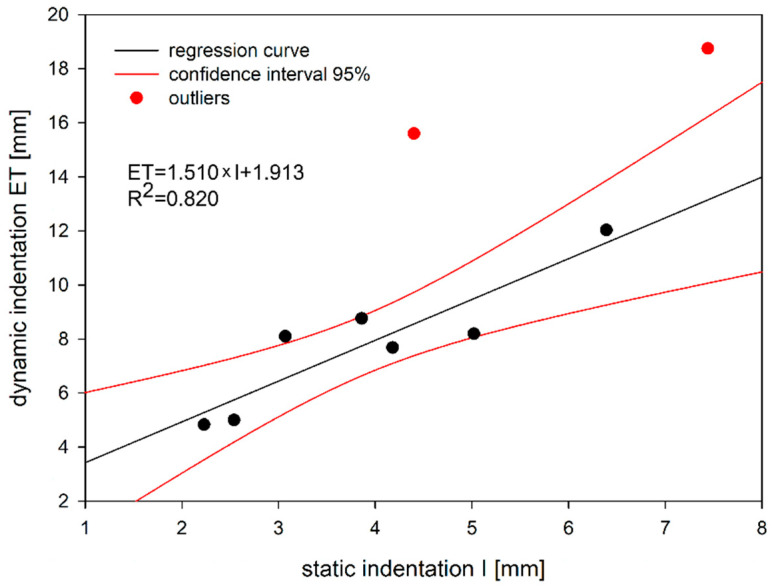
Relationship between static and dynamic indentation.

**Table 1 materials-14-05167-t001:** Basic properties of the 35/50 penetration grade bitumen.

Properties	Unit	Test Methods	Test Results	Requirements According to EN 12591 [27]
Penetration	[×0.1 mm]	EN 1426 [28]	42.8 ± 0.6	35.0–50.0
Softening point	[°C]	EN 1427 [29]	55.1 ± 0.6	50.0–58.0
Fraass breaking point	[°C]	EN 12593 [30]	−13 ± 1.5	≤−5
Density at 25 °C	[kg/m^3^]	EN 15326 [31]	1020 ± 6	no requirements

**Table 2 materials-14-05167-t002:** Basic properties of TLA.

Properties	Unit	Test Methods	Test Results	Requirements According to EN 13108-4 [32]
Penetration	[×0.1 mm]	EN 1426 [28]	4.0 ± 0.5	0.0–4.0
Softening point	[°C]	EN 1427 [29]	101.2 ± 1.0	93.0–99.0
Solubility	[% (m/m)]	EN 12592 [33]	57.6 ± 1.0	52.0–55.0
Density at 25 °C	[kg/m^3^]	EN 15326 [31]	1380 ± 8	1390–1420

**Table 3 materials-14-05167-t003:** Physical properties of TLA-modified binders after RTFOT ageing.

Properties	Unit	35/50 (Base Bitumen)	35/50 +10%TLA	35/50 +20% TLA
Penetration at 25 °C	[×0.1 mm]	31.5 ± 1.1	26.8 ± 1.2	22.3 ± 0.5
Softening point	[°C]	61.7 ± 0.2	65.3 ± 0.3	67.5 ± 0.5

**Table 4 materials-14-05167-t004:** Physical properties of TLA-modified binders after RTFOT ageing.

TLA Addition		ZSV [Pa·s]	
	40 °C	50 °C	60 °C
0% TLA	342,650 ± 35,413	43,690 ± 4314	7703 ± 239
10% TLA	710,933 ± 99,477	78,232 ± 5870	13,312 ± 568
20% TLA	1,172,167 ± 60,001	147,400 ± 5247	24,572 ± 747

**Table 5 materials-14-05167-t005:** Results of regression analysis for static and dynamic indentation tests.

Test Temperature [°C]	Static Indentation Linear Equation (2)		Dynamic Indentation Quadratic Equation (3)	
	R^2^	Standard Error	R^2^	Standard Error
40	0.948	0.453	-----	-----
50	-----	-----	0.902	2.087

## Data Availability

Not applicable.

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
