# Peer review of "The Influence of Zero Shear Viscosity of TLA-Modified Binder and Mastic Composition on the Permanent Deformation Resistance of Mastic Asphalt Mixture"

_materials, 2021, doi:10.3390/ma14185167_

Round 1
Reviewer 1 Report
Dear Authors,
The topic looks interesting, manuscript is well organized but needs following improvement:
1) Introduction seems reasonable but over 50% of the references are older than 2015, so the authors need to bring more recent findings related to the paper topic.
2) Table 1 and 2: Better to put the test method name/standard on the column instead of referencing.
3) Please explain the standard test method or the equivalent one within the preparation section.
4) 3D surfaces are assumed to be in curvature rather than line style. Otherwise, they are not demonstrating/illustrating the findings well.
5) From a theoretical viewpoint, an optimum condition seems to be reachable but from lab test results, it's quite difficult to choose an optimum point as a conclusion.
6) English needs to be checked by a native technical english speaker.
Author Response
Attached pdf file.

Reviewer 2 Report
The reviewer would like to thank authors for their efforts. This is an interesting article. The paper is well-written and the presentation is clear. Test methods are described adequately and clearly. Findings have been presented, analyzed and discussed systematically. My only comment is to have the document edited for minor writing issues. Also, it is recommended to provide limitations of the study.
Author Response
Attached pdf file.

Reviewer 3 Report
This is a very interesting topic. Some comments:
- According to my knowledge, when a material has poor rutting resistance properties, normally the slippage cracking should be considered seriously. This is especially true when poor bonding performance occurs between asphalt materials and bridge structures. How do the authors think about it?
- In this study, the bridge is steel or concrete?
- Line 50: additives, such as wax, in harden bitumen may lead to poor thermal properties. The winter in Poland is quite cold, I donot think this is a good option to modify bitumen in cold regions.
- Figure 1, the black curve is designed or actual?
- Line 220: does 5% still fit the LVE range?
- The analysis and conclusion are reliable. The conclusion is clear.
Author Response
Attached pdf file.
